# Management Barriers to Inter-Organizational Collaboration in Preoperative Treatment of Patients with Hip or Knee Osteoarthritis

**DOI:** 10.3390/healthcare11091280

**Published:** 2023-04-29

**Authors:** Mohsen Hussein, Karmen Erjavec, Nevenka Kregar Velikonja

**Affiliations:** 1Artros Ljubljana, University of Novo Mesto, 1000 Ljubljana, Slovenia; mhussein@artros.si; 2Faculty of Health Sciences, University of Novo Mesto, 8000 Novo Mesto, Slovenia; karmen.erjavec@uni-nm.si

**Keywords:** barriers, inter-organizational collaboration, managerial indecision, hip or knee osteoarthritis

## Abstract

Inter-organizational collaboration among healthcare institutions is widely recognized to improve healthcare services. Because there is a research gap in examining the management barriers to inter-organizational collaboration in countries with less efficient healthcare systems and the reasons for non-implementation of innovations, the aim of this study was to identify key management barriers to inter-organizational collaboration in the preoperative treatment of patients with hip or knee osteoarthritis in Slovenia using a mixed-methods approach with key stakeholders. A cross-sectional study was conducted using multiple methods. An online survey (n = 135) and a multilevel qualitative approach were used, interviewing patients (n = 21), healthcare professionals, and other stakeholders (n = 42). The overall assessment of barriers affecting the integrated approach at the macro, meso, and micro levels revealed that macro-level factors were statistically significantly perceived as the major barriers, while micro-level factors were the minor barriers. There was no significant difference between public and private sector respondents in the barriers at the three levels. However, there were significant differences in the perceptions of different professional groups at the micro and meso levels, but not at the macro level. The analysis of the in-depth interviews confirmed the importance of macro-level barriers. A culture of noncooperation combined with “managerial indecision” prevails in the Slovenian healthcare system due to weak management support for inter-organizational collaboration, with managers and other key stakeholders failing to make urgent decisions due to a lack of autonomy. Interviewees commonly noted that one of the major barriers to inter-organizational collaboration was a lack of resources and staff, particularly of primary care physicians and nurses. In the preoperative treatment of patients with hip or knee osteoarthritis, the culture of non-collaboration and executive indecision are the major macro-level barriers to inter-organizational collaboration in Slovenia.

## 1. Introduction

Inter-organizational collaboration among healthcare settings is widely recognized to improve healthcare services. There are different types of inter-organizational relationships that vary in the intensity of the partner organization’s involvement. Cooperation is the most informal type, while collaboration is more formalized, characterized by a shared vision for exchange, mutually reinforcing activities, continual communication, shared measurement systems, and a coordinating body [1]. Inter-organizational collaboration among different healthcare institutions is a set of processes involving healthcare professionals representing multiple organizations when they work together in patient care [2]. Although inter-organizational collaboration can bring benefits, such as quality improvement, better system efficiency, cost reduction, increased customer satisfaction, and improved access to healthcare, there are many barriers to its implementation in the healthcare system [3,4,5,6,7,8]. As key management activities (planning, organizing, staffing, leading, and controlling) form the basis for implementing inter-organizational collaboration [9], understanding the management barriers to the development of inter-organizational collaboration can help identify and explain the limited efficiency and effectiveness of inter-organizational collaboration in healthcare and highlight the opportunities to improve and promote the implementation of integrated care in such settings [3,7] This is particularly important in the countries with a less efficiently organized healthcare system, such as Slovenia and other comparable Eastern and Southern European countries [10]. In recent years, numerous works in the literature [3] or realist reviews [11,12] on inter-organizational collaboration have been published, of which we refer to Auschra’s framework because it is the most comprehensive, addressing the barriers at all three levels: macro, meso, and micro levels. Auschra’s (2018) systematic literature review identified twenty types of barriers, which were divided into six groups at the macro, meso, and micro levels. At the macro level, there were (i) barriers related to management and regulation and (ii) to financing [3], suggesting that other barriers may be missing, namely a culture of (non)cooperation typical of Slovenia and comparable Central and Eastern European countries [13]. At the meso level, there were (iii) barriers related to the inter-organizational domain, such as lack of leadership and coordination, differences in the design and goals of collaboration, incompatible organizational structures, lack of actors, and power imbalance and conflict; (iv) barriers related to the organizational domain, such as cultural distance, previous experience with collaboration, and organizational vs. collective interests; and (v) barriers related to service delivery, such as lack of mutual understanding, lack of technical standards, lack of communication, differences in professionalization, and resistance to change. At the micro level, there were (vi) barriers related to clinical practice, which include lack of information sharing and confidentiality issues. In this literature review, the barrier “different professionalization” was mentioned most often, followed by “lack of leadership and coordination” and “organizational vs. collective interests.” The author emphasized that this categorization could be biased because inter-organizational collaboration had not been studied in less efficiently organized healthcare systems, such as Slovenia’s. Additionally, there was a lack of studies using a multimethod approach and a lack of those that included all key stakeholders and identified the reasons for collaboration failure. The existing studies included in the review analyzed the barriers of already implemented inter-organizational collaboration [3]. 

Therefore, this study attempted to identify the managerial barriers to inter-organizational collaboration in healthcare in Slovenia, which has not yet been implemented but is successful in most EU countries, e.g., integrated clinical pathways for preoperative treatment of patients with hip or knee osteoarthritis [14,15,16]. Evidence-based integrated clinical pathways involving different professional groups, who are also involved in the conservative treatment of patients with osteoarthritis, has been shown to improve health-related quality of life in these patients [14,15,16]. This clinical indication was selected, as articular cartilage injury is one of the most common and debilitating musculoskeletal conditions [17]. More than 500 million people worldwide suffer from the symptoms of osteoarthritis [18]. The prevalence of knee osteoarthritis, ranging from mild chondrosis to severe joint disease, is present in 60–70% of adults aged 65 years and older [19]. On the other hand, direct medical costs associated with osteoarthritis are higher in younger than in older patients [17]. In the United States, more than one million total knee replacements are performed annually due to osteoarthritis. Longer life expectancy will further increase the number of patients with osteoarthritis, as projections show that this population will nearly double by 2030 [17].

Since inter-organizational collaboration requires different professional groups to work together, it is important to understand how each key group perceives managerial barriers to inter-organizational collaboration. There is no clear evidence in the literature of how different professional groups perceive these barriers [20]. Therefore, the question arises as to how the professional groups responsible for treating patients with hip or knee osteoarthritis and/or referring patients to other professional groups/services within the healthcare system, i.e., primary care physicians, specialists, and physiotherapists, perceive the management barriers to implementing integrated clinical pathways in the preoperative care of patients.

Although in Slovenia—a post-socialist country with a population of 2.1 million and a centralized healthcare system with compulsory social insurance—healthcare is provided mainly by public healthcare centers, the question arises whether the perceptions of barriers to inter-organizational collaboration vary by sector, as the private sector tends to be more flexible and willing to collaborate more to achieve greater operational efficiency [21]. The Slovenian health and social care system, like other Central and Eastern European health care systems (e.g., Croatian, Czech, Slovak, and Polish) underwent significant changes during the transition period. The barriers to inter-organizational collaboration have been identified only in the countries with efficient healthcare systems where inter-organizational collaboration has already been implemented. 

The aim of this study was to identify the main management barriers to inter-organizational collaboration in the preoperative treatment of patients with hip or knee osteoarthritis in Slovenia, a country with a less efficient healthcare system, using a mixed methods approach and engaging key stakeholders in the research. We hypothesized that management barriers to inter-organizational collaboration in the preoperative treatment of patients with osteoarthritis vary significantly (a) by levels (macro, meso, and micro), (b) by professional groups (general practitioners, specialists, and physiotherapists), and (c) by sectors (private and public).

## 2. Materials and Methods

To obtain a comprehensive overview of Slovenian inter-organizational collaboration in the preoperative management of patients with hip or knee osteoarthritis and to identify the main management barriers hindering its implementation, we conducted a cross-sectional study using multiple methods. We first used a quantitative survey, which gave us a broad overview based on statistical analysis, and through qualitative research, we obtained more detailed and emotionally driven insights on detected issues.

### 2.1. Quantitative Approach

Data were collected by an online survey to identify macro-, meso-, and micro-level barriers to inter-organizational collaboration in the preoperative treatment of patients with hip or knee osteoarthritis. Main professional groups involved in the preoperative management of patients with hip or knee osteoarthritis [16] were invited to participate in the study, namely physicians (general practitioners and specialists, mainly orthopedic surgeons working in prosthetics) and physiotherapists, in various Slovenian healthcare facilities. Email addresses of orthopaedic surgeons were obtained from public websites. To obtain relevant groups of general practitioners and physiotherapists, we targeted the group of those who have attended a symposium on prosthetic treatment and are presumably involved in and familiar with the field. The participants were invited by email with a link to the online survey, which included a brief description of the research purpose and objectives. The participants were informed that they agreed to participate in the study by completing the questionnaire. Three reminders were sent one week apart to increase participation. The study was performed between September and November 2022; 135 questionnaires were fully completed (Table 1).

To develop the study instrument, we adapted the measures for all study variables from previously published studies [3,9,22]. To further refine the measurement items from the study construct, we conducted interviews with 5 healthcare scientists and 5 healthcare workers with experience in inter-organizational collaboration. On the basis of the listed papers and interviews, we defined factors potentially influencing interorganizational collaboration for integrated approach, namely 4 factors at the macro level, 14 factors at the meso level, and 3 factors at the micro level (all selected factors are listed in Table 3). Respondents were asked to evaluate the factors from 1 to 5, where 1—“the factor represents a major barrier”, 2—“factor represents a barrier”, 3—“neutral influence”, 4—“supportive factor”, and 5—“highly supportive factor”. We first conducted a pilot study with a sample of 25 health professionals. This part of the questionnaire exhibited a very high degree of internal consistency (Cronbach’s alpha was 0.845). The second part, which included socio-demographic variables, contained five questions on gender, age, educational level, profession, and type of organization (private/public). 

Descriptive analysis and non-parametric statistics were used for presentation of data. The Related-Samples Wilcoxon Signed Rank Test was used for comparison of general assessment of factors at the macro, meso, and micro levels. The Kruskal–Wallis H test (one-way ANOVA on ranks) was used for testing whether there was a statistically significant difference in the assessment of inter-organizational collaboration among the three professional groups of respondents, and the Mann–Whitney U test was used for testing the difference between respondents from the public and private sector. A *p*-value < 0.05 was considered statistically significant. The data were analyzed with SPSS, version 25.0 (IBM Corp, Armonk, NY, USA). 

### 2.2. Qualitative Study

To obtain a comprehensive and in-depth view of the management barriers to inter-organizational collaboration in the preoperative treatment of patients with hip or knee osteoarthritis in Slovenia, a multi-level qualitative approach was used based on the WHO’s definition of all key stakeholders [23]: patients (n = 21), health professionals, and other stakeholders (n = 42) were interviewed to obtain insight at the micro level; the community and other healthcare organizations at the meso level; and the national level professionals (regulatory, financial, professional, and scientific stakeholders) at the macro level. For health professionals, the inclusion criterion was employment in the primary healthcare sector, secondary public and private health organizations, or rehabilitation centres. They were selected by participating in the preoperative treatment of patients with OA in Slovenia, and 63 responded. At the meso level, participants were selected on the basis of their leading role in the organizations (7 managers of healthcare organizations), and at the macro level, on the basis of their role in the healthcare system (5 stakeholders from regulatory, financial, professional, and scientific sectors). At the micro level, the inclusion criterion for patients with hip or knee osteoarthritis was the ability to communicate verbally; for health care professionals, the inclusion criterion was employment in the primary healthcare sector, secondary public and private healthcare organizations, or rehabilitation centers. The study ultimately included 11 physicians–specialists (7 orthopedists, 2 anesthetists, and 2 physiatrists), 7 general practitioners, 4 nurses, 5 physiotherapists, and 3 other health professionals (a dietician, an occupational therapist, and a psychologist). 

A thematic interview guide was developed for data collection on the basis of the management barriers to inter-organizational collaboration found in the literature reviews and on contextual knowledge. The main thematic question was: What are the macro (system), meso (healthcare facilities), and micro (service delivery) managerial barriers to inter-organizational collaboration in the preoperative treatment of patients with hip or knee osteoarthritis?

A thematic interview guide was also developed for data collection on the basis of the barriers to inter-organizational collaboration found in the literature reviews and on contextual knowledge. The main themes were the assessment of the performance of inter-organizational collaboration and the barriers to the implementation of the integrated clinical pathway of preoperative management of patients with hip or knee osteoarthritis. As the data collection progressed, further relevant key information was identified by a cascading effect and added to the list of participants until saturation was reached (no new information was added). All interviews were conducted in person, or online due to COVID-19 pandemic measures, by six experienced researchers. The in-depth interviews, which lasted, on average, approximately 60 min, were recorded with the prior consent of the participants. Anonymized statements were transcribed. 

The data were analyzed using thematic analysis. To determine when data saturation occurred, the thematic analysis was conducted in an iterative cycle simultaneously with data collection. In the first step, the transcribed texts were usually read several times, and descriptive notes were made on the content. Then, a second reading was carried out to code the data, i.e., to mark phrases or sentences and add shorthand or codes describing their content. When patterns were identified among the codes, similar codes were combined to generate a theme. We reviewed the themes by returning to the transcribed texts and checking whether the themes represented the content. The themes were concise and easy-to-understand names based on the barriers identified by Auschra [3] in her systematic literature review of barriers to the integration of care in inter-organizational settings. The analysis of each interview was conducted by two independent researchers. Any problems with data analysis and coding were discussed by the steering committee and resolved by consensus. 

### 2.3. Ethical Issues

Data collection was part of the project ‘Impact of integrated clinical pathways on patient outcomes, communication and cost-effectiveness’ funded by the Slovenian Research Agency (No. L7-2631-3824-2020). The research was approved by the National Committee of Medical Ethics of the Republic of Slovenia (No. 0120-189/2021/3). 

## 3. Results

### 3.1. Results of Quantitative Study 

In the quantitative analysis, the perception of the barriers at the macro, meso, and micro levels was analyzed according to different professional groups and sectors. 

The overall evaluation of barriers, influencing the integrated approach at macro, meso, and micro levels revealed that the factors at the macro level were perceived as major barriers, whereas the influence of factors at the micro level were scored close to neutral in terms of integrated approach. Paired *t*-tests revealed a significant difference between average scores, meaning that the respondents consistently perceived a negative influence of these factors in the same order (Table 2).

Comparison of different professional groups revealed that all groups perceived the factors at the macro level as major barriers, with no significant difference among professional groups. The differences appeared significant at the meso and the micro levels, where the listed factors were, in general, perceived as major barriers by specialist physicians, while physiotherapists were the least critical. However, general practitioners perceived ‘(non)communication between actors’ and ‘patient requests for additional treatment/referral’ as greater barriers in comparison with other two professional groups (Table 3). 

There was no significant difference at the three levels between respondents from the public and the private sectors regarding the barriers. However, the respondents from the private sector were, in general, less critical in the estimation of the negative influence of individual parameters. This difference appeared significant in the estimation of the influence of management, which was the highest scored factor in the private sector, revealing the perceived positive influence of the management on an integrated approach, whereas the neutral role of the management was estimated by respondents from the public sector (Table 3).

### 3.2. Results of Qualitative Study 

In the qualitative analysis of the study, the reasons for the lack of inter-organizational collaboration in the implementation of an integrated clinical pathway for the preoperative treatment of patients with hip or knee osteoarthritis were examined in greater detail (Table 4).

#### 3.2.1. Macro-Level Barriers 

All interviewees indicated that the main managerial barrier to inter-organizational collaboration occurred at the macro level. It is the systemic barrier, due to a less efficient healthcare system. All groups interviewed pointed out that the system was ineffective due to a lack of resources and excessive administrative work. A typical statement was from Patient 16: 

“Our healthcare system does not work well because not enough money is provided and at the same time too much bureaucratic work has been imposed on doctors and nurses. There is a lack of funds for sufficient medical personnel, for sufficient treatments, for investments. … Of course, some people are making big profits. But for us all this is not enough, and then they ordered the doctors and nurses to fill in meaningless administration. Yes, I can see while I am waiting for treatment. So, there is lack of money, no time for patients, and no will to introduce something as good as this preoperative treatment.”

Most healthcare professionals and other stakeholders at the national level pointed primarily to the culture of noncollaboration, which is widely believed to be the result of power imbalances and conflicts in which politicians, financiers, and other key stakeholders have too much power in decision-making in public healthcare institutions and government healthcare organizations (e.g., the Ministry of Health and the National Institute of Public Health). As a result, managers and other key decision makers at these institutions fear that their decisions will not be approved and, thus, become obsolete. Therefore, due to a lack of autonomy and fear of being replaced, they do not make strategic decisions for creating the conditions for inter-organizational collaboration and make other important changes (especially regarding investments and personnel). Therefore, they prefer not to decide. An example of a typical answer is:

“In my opinion, the general problem is the indecision of the main players. Why? Because they are afraid of being replaced if they do not play the way others want them to. So, the main problem is that public healthcare in general is inefficient and disorganized, with politicians, the insurance company, and some other institutions such as the medical association and the unions taking the lead, each defending its own interests. I know that we have many cases where good innovations are ready, but these people are afraid of being replaced if they do not make the decision expected of them, so they leave the documents in the drawer embark on this integrated path, and last but not least, to make all the essential changes, it is an imperative to change the system of organization, management, and administration.” (Stakeholder 2).

#### 3.2.2. Meso-Level Barriers

Poor management of healthcare facilities has been identified as one of the main reasons why inter-organizational collaboration does not work adequately. Interviewed patients pointed to the poor organization in general, whereas healthcare professionals and other stakeholders repeated the problem of the influence of politics and other actors on decision-making in healthcare facilities, such as hospitals, healthcare centers, and nursing homes:

“The main problem is poor management. I can say that out loud, even though I am the director of a healthcare center. Because of the politicization of healthcare, I do not have the autonomy to make decisions. And the losers of all of this are the citizens who years ago had difficulty accessing specialists, but today they have difficulty accessing a general practitioner, which has never existed before. This is a violation of constitutional and human rights. It is necessary to change the management of public healthcare facilities and give them the necessary autonomy in administration by appointing appropriate experts with knowledge in healthcare and healthcare management. The facilities also need effective supervisory boards, which must be composed of healthcare professionals and not be politically motivated. Again, a depoliticization of the healthcare system is needed. The role of politics is simply to create the conditions for the optimal functioning of the healthcare system. The healthcare system should be managed by experts who are most knowledgeable in this area.” (Stakeholder 3)

All interviewees stressed that the biggest management barrier in practice was the lack of staff, as Nurse 4 said, “We have a big shortage of GPs and nurses. The low pay for so much work is forcing my colleagues to work elsewhere.” Some interviewees also noted that insufficient digitization of the healthcare system and incompatible ICT were major problems of implementing preoperative treatment of patients with hip or knee osteoarthritis. Or, as Specialist 6 said, “How are we supposed to collaborate with different organizations when each has its own ICT system. We also have different ICTs in the hospital. Absurd! And this is where we could really save money so that there is no duplication of effort! I’m really angry!”

#### 3.2.3. Micro-Level Barriers

Lack of communication is another barrier at the micromanager level due to the lack of time, incompetence, or personal characteristics: “I know that the problem of implementing this pathway may be due to the lack of communication. We do not communicate enough with professionals from other institutions because we just do not have time. But I must admit that sometimes I prefer to avoid conflicts by not communicating at all.” (Specialist 5). Or, as Patient 14 said, “They do not have time to do anything else because they can barely keep up!”. Healthcare professionals and stakeholders also emphasized that “there is often a lack of willingness to communicate about innovations, or about anything important or anything that might cause conflicts or additional work” (Stakeholder 4), such as the introduction of preoperative treatment of patients with hip or knee osteoarthritis. Some health professionals and stakeholders also pointed to the lack of staff motivation as a reason for not adopting innovations. A typical statement came from Specialist 4, who emphasized an extremely individualistic viewpoint: “I do not know why we should introduce this? What do I get out of it?”

## 4. Discussion

The use of multiple methods provided a broader and deeper insight into management barriers to inter-organizational collaboration in the integrated clinical pathways in the preoperative treatment of patients with hip or knee osteoarthritis. On the basis of the homogeneity and content of the arguments, the analysis of the statements of the interviewed healthcare professionals and other key stakeholders reveal that inter-organizational collaboration in the health system is generally weak and not limited to the integrated pathway under study only.

The next main finding is the importance of macro-level barriers (noncollaboration with managerial indecision), which manifests itself in several ways. Firstly, the results of both the survey and the in-depth interview show that the biggest barriers are at the macro level. Secondly, the survey results show that among the three levels of barriers, respondents emphasize the macro-level barriers. Thirdly, all professional groups agree on the importance of macro barriers, as there are no statistically significant differences among them. Fourthly, interviewees’ reference to the same barriers to inter-organizational collaboration at both the macro and the meso levels points to a major problem regarding the influence of politicians, financiers, and other influential actors on healthcare managers, leading to diminished autonomy of key decision-makers and their indecision, which, in turn, results in a less efficient healthcare system because key reforms have not progressed. The decision-making autonomy of healthcare managers is very important, as limited decision-making autonomy is negatively associated with their job performance as managers [24]. 

Several studies have already revealed that in Slovenia and other post-socialist countries of Central and Eastern Europe, there are major difficulties in decision-making by healthcare managers at different levels. In explaining this phenomenon, we considered the studies of “decision pathologies” that refer to situations in pluralistic environments, with multiple actors expending much energy in the long run without taking concrete strategic action [25,26]. Our findings contribute to the studies of decision pathologies by showing that “managerial indecision” occurs in a situation where decision makers do not make urgent strategic decisions because they fear losing their positions due to the lack of autonomy and often unwarranted criticism from influenced people/organizations or because they make decisions that contradict the expectations of influenced people/organizations. As a result, they prefer not to make decisions that would require either independent decisions or much coordination with supervisory structures. In fact, due to the postponement of the much-needed healthcare reform, Slovenia has not introduced a modern management system into the prevailing public healthcare system. It is evident that politicians and other interest groups (e.g., medical associations, unions, and financial and other lobbyists) have a strong influence on public healthcare management [10,27,28,29,30]. This limits managers’ decision-making processes, as managers are often very hesitant to introduce strategic changes in the face of frequent superficial criticism [31,32]. The result is indecision among managers and other key actors and, consequently, weak adoption of healthcare innovation. 

Poor organization of healthcare institutions in Slovenia can also be attributed to the fact that most managers of healthcare institutions are physicians who were good professionals but do not have enough management knowledge and experience; thus, the system loses a good physician and gains a bad manager [32]. In addition, managers of healthcare institutions are relatively poorly paid compared with business managers. Therefore, it is not exactly rational to take a job that pays poorly and involves great responsibility, minimal authority, and much pressure [33].

The results show that one of the biggest obstacles is also the lack of resources and personnel, especially of general practitioners and nurses. The shortage of personnel is a major challenge for the Slovenian healthcare system, as outlined in the State of Health in the EU: the number of general practitioners in Slovenia (3.3 per 1000 inhabitants) is significantly lower than the EU average (4.0 per 1000 inhabitants), and although the number of nurses is higher than EU average (10.5 per 1000 inhabitants), the number includes those who have completed vocational training only and do not comply with the Directive on Regulated Health Professions [34]. 

Our findings are consistent with those of Auschra’s literature review on the barriers to integrating care in inter-organizational settings [3]. However, certain barriers were less explicitly mentioned by our respondents and interviewees, as macro-managing barriers predominated. For example, the barriers of incompatible organizational structure, differences in collaboration design and goals, lack of mutual understanding, and organizational vs. collective interests were implicitly expressed as being less important due to a less efficient healthcare system. The fact that macro barriers are important in a less efficient healthcare system is also evidenced by the finding that the power imbalance, which other studies [3] have found at the meso level, manifests itself in Slovenia at both the macro and meso levels. 

The factors at the macro level are perceived as major barriers; however, there are no significant differences among professional groups or between the public and private sector. Healthcare financing is perceived as the greatest barrier by all professional groups and sectors. These issues have persisted in the Slovenian healthcare system for several years [10,27,28,29,30]; thus, healthcare professionals perceive them more explicitly as barriers that are so important or big that all professional groups rate them similarly. Significant differences among professional groups appear in the perception of management barriers at the meso and micro levels, reflecting different roles of investigated professional groups at these levels. Physiotherapists suffer less from staff shortages in comparison with medical doctors, and they perceive less influence of different goals of institutions in public and private sectors, as well as different professions and (non)communications among actors as the barriers to integrated organization of clinical pathways. They are also the least critical about two of the three factors at the micro level, which can be explained, at least in part, by the intermediary role of physiotherapists, as the patients are referred to them by medical doctors, and they are also more involved with patients at the preoperative stage than GP or specialists. The respondents from the private and public sectors are similar in their perceptions of the influence of the listed barriers at all three levels. The only significant, but very important, difference between them is that the respondents from the private sector perceive the management as a most positive factor regarding the integrated approach. This could be explained by the fact that the private sector tends to be more flexible and prone to collaborate in order to achieve greater operational efficiency [21]. The use of integrated resources is a prerequisite for achieving efficiency from the patient and from organizational and inter-organizational perspectives. To overcome the barriers to inter-organizational collaboration at the macro, meso, and micro levels, recommendations for specific steps at all levels have been developed within the framework of change and transformation management using a systemic approach, cooperation (vertical integration), and horizontal, functional, and normative integration, including various changes from legal to behavioral elements [35,36]. At the macro level, inter-organizational collaboration and other important innovations in healthcare in Slovenia and comparable Central and Eastern European countries are possible only when the most important macro-level barriers are removed: politicization of healthcare decision-making and diminished management autonomy. This may be achieved by changing the legislation to appoint suitable professionals with knowledge in both healthcare and healthcare management, working as independently as possible from daily politics, improving the financing of the healthcare sector, and, consequently, employing enough healthcare professionals. Currently, most of these factors in Slovenia differ significantly from those in other OECD countries and have been highlighted as possible causes of less efficient organization of the healthcare system [10,27,28,29,30]. At the macro level, it is also necessary to implement a compatible digital system that will enable the exchange of documentation among different organizations. To change the dominant culture of non-collaboration, key authorities in the healthcare sector should encourage inter-organizational collaboration. At the meso level, systematic inter-organizational cooperation in the Slovenian healthcare system requires coordination, communication, and common goals among healthcare organizations. Strategies that can be used to improve collaboration among organizations include enabling the selection of appropriate leaders for healthcare facilities who are knowledgeable in both healthcare and healthcare management; establishing common goals that all organizations will look forward to achieving; developing common strategies, policies, and protocols that all organizations will follow, e.g., establishing common care pathways (e.g., the clinical pathway for preoperative treatment of patients with hip or knee osteoarthritis) that will define the roles and responsibilities of all organizations involved in patient care; establishing a governance structure to support collaboration and communication among organizations; promoting the sharing of resources, such as staff, equipment, and data; encouraging regular communication among organizations to facilitate coordination and problem solving; and encouraging and financially rewarding organizations that work together, e.g., paying well for cases where a broader council of different organizations is involved in the decision on treatment. At the micro level, healthcare professionals should work together to coordinate care for individual patients. Strategies that will help improve inter-organizational collaboration at the micro level include establishing regular communication channels among providers involved in patient care, e.g., via electronic health records or secure messaging platforms; developing shared care plans to outline the goals of care for each patient; and encouraging professionals to work together to address issues or challenges that arise during the care process. At the micro level, healthcare professionals should also be encouraged to participate in decision-making processes to become more aware of and more motivated for innovative treatment.

Although the findings have at least partially filled the research gap on the management barriers to inter-organizational collaboration in Slovenia and in comparable Eastern and Southern European countries, the main limitation of the study is the specific situation during and after the COVID-19 pandemic, when management decisions were particularly limited or aimed at regulating the situation caused by the pandemic. As the sample was specifically selected according to the work with preoperative treatment of specific group of patients, the representativity of the sample according to demographic characteristics could not be verified; therefore potential effects/bias of the sample’s characteristics on the results could not be excluded.

## 5. Conclusions

In the preoperative treatment of patients with hip or knee osteoarthritis, the culture of non-collaboration and executive indecision are the major macro-level barriers to inter-organizational collaboration in Slovenia. One of the biggest obstacles to inter-organizational collaboration is also the lack of resources and personnel, especially of primary care physicians and nurses. 

## Figures and Tables

**Table 1 healthcare-11-01280-t001:** Sociodemographic characteristics of the sample (n = 135).

Demographic Parameter	Category	No. of Respondents n (%)
**Gender**	Female Male	87 (64.4%) 48 (35.6%)
**Age**	<30 years	17 (12.6%)
	31–40 years	22 (16.3%)
	41–50 years	44 (32.6%)
	51–60 years	32 (23.7%)
	61 years <	20 (14.8%)
**Profession**	MD–general	55 (40.7%)
	MD–specialist	39 (28.9%)
	Physiotherapist	41 (30.4%)
**Type of organization**	Public sector Private sector	84 (62.2%) 51 (37.8%)

**Table 2 healthcare-11-01280-t002:** Comparison of perception of management barriers to inter-organizational collaboration at different levels (n = 135).

Organizational Level	Median	Mean	SD
Macro level	2.00	2.14	0.61
Meso level	2.50	2.50	0.54
Micro level	3.00	2.82	0.70
Related-Samples Wilcoxon Signed Rank Test			**Sig. (P)**
Macro vs. meso			<0.001
Meso vs. micro			<0.001
Macro vs. micro			<0.001

**Table 3 healthcare-11-01280-t003:** Perception of management barriers to inter-organizational collaboration at the macro, meso, and micro levels by different professional groups and by type of organization (n = 135).

	All Responders		Group of Professional				Type of Organization		
Estimation of the Influence of the Following Factors on Interorganizational Collaboration for Integrated Approach:	Median	Mean *(SD)*	MD–General (Mean)	MD–Specialist (Mean)	PHT (Mean)	Kruskal–Wallis H Test *(P)*	PUB (Mean)	PRIV (Mean)	Mann–Whitney U Test *(P)*
**n**	135		55	39	41		93	52	
Macro level—average	2.00	*2.14 (1.04)*	2.15	2.05	2.09	*0.793*	2.09	2.12	** *0.701* **
The culture of (non)cooperation in Slovenia	2.00	*2.22 (0.87)*	2.22	2.21	2.24	*0.969*	2.24	2.20	*0.901*
Legislation in the field of healthcare	2.00	*2.31 (0.77)*	2.38	2.33	2.20	*0.295*	2.27	2.37	*0.620*
Healthcare financing (e.g., standardization of services and salaries)	2.00	*1.83 (0.77)*	1.93	1.64	1.88	*0.522*	1.80	1.88	*0.531*
Control of work by the insurance agency (ZZZS)	2.00	*2.13 (0.84)*	2.15	2.26	2.03	*0.309*	2.14	2.11	*0.884*
Meso level—average	2.50	*2.55 (0.54)*	2.50	2.44	2.74	*0.008 **	2.55	2.57	*0.949*
Management (coordination of work by management)	3.00	*3.24 (0.90)*	3.40	3.18	3.10	*0.241*	3.05	3.57	*0.01 **
Differences in the understanding of cooperation and cooperation goals	3.00	*2.75 (0.69)*	2.75	2.56	2.93	*0.042 **	2.73	2.78	** *0.708* **
Different organization of institutions (different working hours, hierarchy, etc.)	3.00	*2.54 (0.67)*	2.56	2.41	2.63	*0.299*	2.55	2.53	*0.854*
Power positions and conflicts among institutions	2.00	*2.37 (0.72)*	2.29	2.26	2.59	*0.025 **	2.38	2.35	*0.868*
Staff size/number	2.00	*2.21 (0.99)*	2.04	2.00	2.66	*<0.001 ***	2.14	2.33	*0.398*
Experience from previous collaborations	3.00	*3.14 (0.97)*	3.05	3.08	3.32	*0.204*	3.18	3.08	*0.477*
Differences in organizational culture (way of working norms, values)	3.00	*2.59 (0.83)*	2.49	2.56	2.76	*0.099*	2.54	2.69	*0.486*
Different interests and goals of individual institutions (public/private)	2.00	*2.49 (0.84)*	2.40	2.33	2.76	*0.012 **	2.51	2.45	*0.749*
(Mis)understanding the role of other organizations	2.00	*2.33 (0.66)*	2.29	2.33	2.39	*0.620*	2.36	2.29	*0.607*
Technical standards	3.00	*2.71 (0.75)*	2.64	2.69	2.83	*0.218*	2.75	2.65	*0.443*
(Dis)trust among actors	2.00	*2.41 (0.80)*	2.40	2.26	2.59	*0.084*	2.51	2.25	*0.085*
(Non)communication among actors	2.00	*2.11 (0.94)*	1.89	2.15	2.37	*0.031 **	2.15	2.04	*0.686*
Different profession or professionalization	3.00	*2.70 (0.99)*	2.71	2.38	3.00	*0.015 **	2.65	2.78	*0.368*
(Un)acceptance of changes/innovations in the organization	2.00	*2.41 (0.95)*	2.38	2.18	2.66	*0.054*	2.36	2.49	*0.377*
Micro level—average	3.00	*2.82 (0.69)*	2.72	2.57	3.18	*<0.001 ***	2.81	2.83	*0.728*
Documentation (e.g., scope, content, and technology)	2.00	*2.41 (0.99)*	2.35	2.03	2.85	*0.001 **	2.31	2.57	*0.107*
Confidentiality between healthcare professional and patient	3.00	*3.40 (0.85)*	3.49	3.08	3.59	*0.120*	3.44	3.33	*0.330*
Patient requests for additional treatment/referral	3.00	*2.77 (0.86)*	2.38	2.96	3.13	*0.001 **	2.83	2.68	*0.551*

Legend: SD—standard deviation, N—number of respondents, MD—medical doctor, PHT—physiotherapists, PUB—public sector, PRIV—private sector. * The difference in distribution is significant at the 0.05 level. ** The difference in distribution is significant at the 0.01 level.

**Table 4 healthcare-11-01280-t004:** Management barriers to inter-organizational collaboration in preoperative treatment of a patient with hip or knee osteoarthritis.

Barriers	Patients	Health Professionals	Other Stakeholders
**Macro level (system)** Insufficiently efficient system due to culture of non-collaboration			
	Insufficiently efficient system		
		Power imbalance and conflicts: overpowering politicians and financier Indecision of key actors	Power imbalance and conflicts: overpowering politicians and financier Indecision of key actors
Administrative barriers Funding barriers	Too much administration	Too much administration	
	Lack of resources	Unpaid inter-organizational collaboration	Lack of resources
**Meso level (healthcare setting)** Poor management Lack of technological standards Lack of staff			
	Disorganization	Power imbalance and conflicts: overpowering politicians and financier Indecision of key actors	Power imbalance and conflicts: overpowering politicians and financier Indecision of key actors
	-	Incompatible IT infrastructure	Incompatible IT infrastructure
	Shortage of GPs	Shortage of nurses and GPs	Shortage of nurses and GPs
**Micro level (service delivery)** Lack of communication			
	Lack of time for communication	Lack of time for communication, incompetence, or personal characteristics	Lack of time and willingness for communication

## Data Availability

The data presented in this study are available on request from the corresponding author.

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
