# Peer review of "Management Barriers to Inter-Organizational Collaboration in Preoperative Treatment of Patients with Hip or Knee Osteoarthritis"

_healthcare, 2023, doi:10.3390/healthcare11091280_

Round 1

Reviewer 1 Report

The authors conducted a cross-sectional study to identify the main management barriers to interorganizational collaboration in the preoperative treatment of patients with hip or knee osteoarthritis in Slovenia. The statistical methodology has many critical issues. The objective of the study and the questionnaire used to measure the primary objective are not well described. For intergroup comparisons, it is not clear when the comparisons are paired and independent. the Nonparametric Statistical Methods stated in the methods are not described in the results. The study needs to be revised methodologically.

Reviewer 2 Report

The authors touched on an important topic with a thoroughly designed and presented study. The questions are clearly presented, the methods are fine, and the discussion is rich. Overall I see potential in this study. With that said, I have two minor questions and hope to see the authors' response:

1. On page 2, the authors used Auchra's literature review to begin the discussion on the three-level barriers. I wonder is there any other work related to the identification/discussion of such barriers? If yes, the authors should justify the merit of Auchra's work and why it is applicable to your study. If not, the authors should also state that, and make attempts to evaluate Auchra's framework (I see the authors have already done this on "different professionalization", but any other?)

2. This study has comprehensively explored the barriers by levels, groups, and sectors, etc. While the issues have been clearly presented and discussed, I wonder if the authors could attempt to make recommendations. With that, the paper will bring more value to practitioners as well. 

Reviewer 3 Report

Introduction

The introduction is too long with some redundancies for instance between line 105 and 111. Some elements would be better in the discussion than in the introduction for instance Line 106 to 108 and Line 109 to 132.

Line 132-134 the statement is not so much related to the whole paragraph.

Methods

Quantitative survey

Retrieving emails on public websites is unconventional and I am wondering how this method guaranty a representative sample for the quantitative statistical analysis. To complete my comment, can the authors detail:

-   how many emails were sent (what is the response rate?)

-    who is the targeted population for this quantitative analysis?

-   is the sample representative of their targeted population (notably with 64% of female)?

The authors should also comment on the potential effects/biais of the sample’s characteristics on their results.

As Likert scale is used in the questionnaire, I would privilege the median as summary parameter (at least when summarizing the individual questions) and I would privilege the use of non-parametric tests all along.

Qualitative interview

It is unclear who the stakeholders are and what is their interests in healthcare coordination in OA.

Did the authors use a specific software for the qualitative analysis?

Results

In Table1 and Table 2 the “all respondents” Macro average differ (2.10 vs 2.14) is that correct? What is the difference between the 2 summaries?

The header of Table 2 is a bit confusing:

- Are the values for the stratified results by professional status and organisation type also means?

- the number of participants (N) could simply be added in parantheses besides the label of the column

Discussion

Can the perception of physiotherapists be explained by their intermediate position/role in health care pathway?  There are referred by other health care professionals to patients and thus are probably more prone not have collaborators (denser network) that GP or specialists. 

What are the strengths/limits of this study?

In the observed quantitative results, can any bias be attributed to the sample's characteristics?

Reviewer 4 Report

Adopting the point of view of healthcare consulting I should like recommend the delineation of concrete steps and actions contributing to the improvement of the status quo. I is useful to refer to the literature of change and transformation management, since obstacles of transformation have to be overcome: legal, technological, economic, behavioral and medical.

Round 2

Reviewer 1 Report

The authors responded to requests for review of statistical methodology . In this form the work can be published